# Effect of Surface Topology on the Apparent Thermal Diffusivity of Thin Samples at LFA Measurements

**DOI:** 10.3390/ma15144755

**Published:** 2022-07-07

**Authors:** Robert Szczepaniak

**Affiliations:** Faculty of Aviation, Polish Air Force University, Dywizjonu 303 Street No 35, 08-521 Deblin, Poland; r.szczepaniak@law.mil.pl

**Keywords:** COMSOL Multiphysics, numerical modeling, heat transfer, rough structures, profilometry

## Abstract

This paper deals with the problem of the influence of surface topography on the results of thermal diffusivity measurements when determined using the instantaneous surface heat source method, also called the pulse method. The analysis was based on numerical tests carried out using Comsol Multiphysics software. The results of experimental investigations on the actual material structure using an electron microscope, an optical microscope and a profilometer were used to develop a numerical model. The influence of the non-uniformity of the surface of the tested sample on the determined values of half-time of the thermal response of the sample’s rough surface to the impulse forcing on the opposing flat surface was determined by developing the data for simulated measurements. The effect of the position of the response data reading area on the obtained simulation results was also analyzed. The obtained results can be used to improve the accuracy of experimental heat transfer studies performed on thin-film engineering structures depending on the uniformity and parallelism of the material applied to engineering structures. The difference in half-life determination error results for various analyzed models can be as high as 16.7%, depending on the surface from which the responses of the heating impulse are read. With an equivalent model in which 10% of the material volume corresponds to the rough part as a single inclusion, hemisphere, the error in determining thermal diffusivity was equal to 3.8%. An increase in the number of inclusions with smaller weight reduces an error in the determination of thermal diffusivity, as presented in the paper.

## 1. Introduction

The process of coating the surface of an object or substrate with a very thin layer is referred to as coating. The layer can be a type of thin polymer sheet, paint, or varnish used for protective/decorative purposes. Most industrial products go through the coating process not only to prevent corrosion, but also to make them more attractive. The coating consists of the development of a thin layer that can be varnished or polymeric on a fabric or substrate, etc. [1].

Novas et al. [2] described the characteristics of the contributions made by scientists around the world in the field of solar coatings between 1957 and 2019. Photovoltaic systems depend greatly on the physical and chemical properties of their materials, the wavelength of the captured light, its intensity and angle of incidence, surface or texture characteristics, and the presence or absence of superficial coatings. In addition to these factors, temperature, pressure, machinability, durability, price, and lifetime cost are important in material selection.

To prevent dangerous failures, Cui et al. [3] developed a thin-film thermocouple temperature sensor to measure the temperature of the rolling elements of the bearing in real time while the train is running, bearing in mind the characteristics of the rapid change in the temperature of the bearings.

He et al. [4] investigated the coating technology of the DLC film for the orbiting screw of scroll compressors in air conditioning systems in order to reduce the friction coefficient and improve film adhesion to the substrate, wear resistance, and compressor efficiency. The friction coefficient of the DLC film is much lower than that of an anodic oxide film, as a result of which the DLC film has a good wear-reducing effect and excellent self-lubricating properties.

Mohammed et al. [5] investigated ZnS thin layers prepared by deposition in a chemical bath. They investigated the structural properties of films deposited and annealed at a temperature of 300 °C. This research found that relatively small grains were randomly formed and the deposited film had a heterogeneous surface with some cracks. Additionally, it was observed that the annealed samples had higher content of Zn than S, and the white color of the deposited thin layers did not change after annealing.

Poddighe et al. [6] addressed the problem of producing hydrophobic thin films from the liquid phase. Interest in the production of hydrophobic surfaces is constantly growing due to their wide application in several industries. Thin liquid-phase layers can be deposited on a variety of surfaces using a wide variety of techniques, and the design of the precursor solution offers the possibility to fine-tune the hydrophobic properties of the coating layers. The general trend is to design multi-functional films that have different properties in addition to being hydrophobic. The control of surface wettability is a key technological issue in several areas, such as microelectronics, separation membranes, car windows, self-cleaning surfaces, motion indicators, and biotechnology [7,8,9,10,11,12]. The leading technologies used for hydrophobic/hydrophilic thin film deposition and surface modification include chemical vapor deposition [13], laser ablation [14,15], and plasma treatment [16]. Superhydrophobic surfaces have a variety of uses in self-cleaning, anti-icing and anti-stick applications. Haj Ibrahim et al. [17] investigated the effect of surface topography on hydrophobic coatings both numerically and experimentally. Profilometry was used to create a numerical representation of the surface.

In addition to physical deposition techniques, the wet chemistry route, in particular sol–gel processing [18], has also met with great interest in the production of highly controlled hydrophobic-hydrophilic systems. Li et al. [19] proposed and improved an easy and cost-effective method involving acid etching and stearic acid self-assembly to successfully produce Al-based superhydrophobic fin tube heat exchangers. The 3D topography analysis showed suitable micro-nanostructures, whereas XPS and FTIR spectra showed chain self-assembly, essential for the realization of a superhydrophobic surface.

For temperature measurement, Dong et al. [20] produced a weak thin AlN layer with numerous defects. It was found that the thin film diffraction peak (002) increased monotonously with an increasing annealing temperature and an annealing time. This phenomenon is attributed to the evolution of defects in the AlN film network. Therefore, the relationship between defects and annealing can be expressed by the shift (002) of the diffraction peak, which can be used for temperature measurement. In addition, an algorithm for temperature interpretation was established, and software for temperature interpretation using MATLAB was also built. The temperature interpretation is realized by the software with a relative error of less than 7%. This study is of great importance in promoting the accurate measurement of the temperature at the surface of the high-temperature component. An accurate measurement of the surface temperature of a turbine blade and the control of its temperature distribution is an important basis for diagnosing turbine blade failure. The methods of measuring the surface temperature of turbine blades mainly include thin-film thermocouples [21,22,23], temperature indicating paint [24], infrared radiation [25,26,27], and irradiation crystals [28,29,30]. Based on the advantages of irradiating crystals, the authors [20] proposed an easy and cheap method of temperature measurement using a thin-film crystal. It was noted that the crystal quality could be strengthened after annealing [31,32,33], indicating that the AlN thin layer is a promising material for temperature measurement.

On the other hand, the thickness of the coating dramatically affects the functionality of the coatings. Accordingly, techniques used to determine thickness are of great importance for research and technology related to coatings. The use of thin films has become ubiquitous in many areas of the science and industry sectors. Coatings are widely used to obtain a synergistic effect between the characteristics of the substrate and the coating material. They can improve the physical, chemical, and aesthetic properties and lower the cost of the final product. For all these reasons, thickness measurement in composite materials is mandatory, both for obtaining the desired properties in the final component, and in order to keep costs under control [34,35].

The aim of this work is to address the problem of heat transfer in unevenly shaped samples based on previously published experimental studies in a temperature-sensitive paint (TSP) material [36]. It also deals with ongoing research on pressure-sensitive paint (PSP), which is dedicated to wind tunnel testing [37,38]. Due to the unproblematic application of this paint on surfaces, it is also used for industrial and scientific measurements [39,40,41]. The use of paint, particularly in wind tunnels, involves measurements that are quick-changing but with not much variation in the temperature measurement range; therefore, the authors of article [36] presented the thermophysical properties of the paint used in experimental studies in this field. However, during the measurements, the authors observed a large influence of the roughness of the prepared test samples on the measurement of thermal diffusivity determined using the classical method [42,43,44], as well as when using the modified method [45,46,47], especially of the sputtered or applied surface opposite to the flat substrate, similarly to Refs. [36,48,49,50].

In the case of heat transfer studies based on the determination of the global surface temperature distribution, it is necessary to observe, in the analysis, the thermal properties of all layers [36], as well as the very topography of the surface opposite to the area of influence of the laser pulse. Therefore, the thermal properties of the layers should be known as accurately as possible, as should the surface topography itself. TSP, similar to PSP, consists of molecular sensors embedded in a binder, usually polymer-based, which can affect the production of samples with a small cross-sectional dimension that automatically implies the problem of obtaining perfectly flat sample surfaces for experimental testing.

The determination of surface topography is particularly important when studying thin-film structures and the physical properties of materials whose property depends on the thickness of the sample (in particular, if the thickness is a square function, as is the case of determining thermal diffusivity by the laser flash method (LFA—Laser Flash Analysis), where the thickness is to the second power). Therefore, the determination of the thermal diffusivity of materials aerosolized on surfaces is not straightforward and is subject to high methodological error. By using an optical profilometer, it is possible to determine the topography and, at a later stage, to include it in numerical simulations to determine the accuracy of the thermal diffusivity parameter.

The use of numerical calculations by means of commercial software is increasingly popular, especially in mechanical engineering [51,52]. Comsol Multiphysics software is very frequently used for calculations in the area of heat transfer, as well as for the multi-physics simulation platform [53,54]. Recently it has also been particularly popular for heat transfer calculations in thin-film structures [55,56,57,58,59,60,61,62]. The influence of waviness is of great importance in calculations of heat transfer phenomena in thin films and in complex flow models used in, among other things, aviation technology, e.g., rocket nozzles [63], and in heat conduction itself in microchannel flows [64], which is a frequently tackled problem of modern mechanical engineering.

This study attempts to investigate the thermophysical properties of the Temperature Sensitive Paint (TSP) coating applied by spraying with an airbrush using numerical tests of substitute models. The error in determining the half-life was determined, and directly translates into the error in determining the thermal diffusivity of the coating.

## 2. Subject of Numerical Research

In the research presented here, a numerical analysis of heat transfer was carried out in models corresponding approximately to the structure of paint sprayed on a flat surface using UniTemp temperature-sensitive paint, manufactured by Innovative Scientific Solutions Incorporated (ISSI, Dayton, OH, USA). The thickness of the prepared model and the surface topography were determined to be massive and 90:10 (flat part of the sample: upper part—irregularities on the sample), which was simulated as inclusions in the form of hemispheres. Because of the problem of surface accuracy, the surfaces of this layer were checked using an optical profilometer (Figure 1a,b) and an electron microscope (Figure 1e,f). For testing purposes (for numerical calculations), a unit layer and maximum thickness deviations of approximately 10% were modeled (one variant of up to 20% was also made for checking). To illustrate the maximum deviation from parallelism of the sample, the roughness of the sputtered material is shown (Figure 1c,d). After a visual analysis, several geometric models were prepared. They are presented later in this article. The samples were modeled in such a manner that on one side there was a flat surface (in line with real-world testing), while on the opposite side, the surface was non-parallel, which corresponds to the replacement model of structure non-uniformity in real-world testing [36].

For the empirical studies, several samples were made with very different surfaces. In one batch of test samples for real-world testing, the coating thickness was not uniform and varied from 27 to 32 micrometers (Figure 1e,f). The inclusions stood out significantly, as can be seen in the electron microscope images. The densities and dimensions of the geometric TSP carriers were determined from the representative coating thickness. In a further batch of samples, the surface roughness of the TSP layer was examined and characterized by 2D and 3D surface scanning, using an FRT MicroProf 100 (FRT GmbH, Bergisch-Gladbach, Germany) optical profilometer. The morphology of the 3D surface is shown in Figure 2a. Figure 2b shows the results of the surface mapping. The surface profile along the red line in Figure 1c is shown in Figure 1d. The spectrum average is 45 µm, while 80% of the data points are within ±15 µm of this value. 

Because the original material was applied to a flat substrate, one surface of the material created for the simulation is flat (the effect of the laser pulse causing the temperature rise will be simulated on this surface). A numerical simulation of impulsive forcing, applied to the determination of thermal diffusivity, such as in [65], will make it possible to assess the applicability of the finite element method to the influence of the parallelism of the surface of the material under investigation.

## 3. Methodology for Determining Thermal Diffusivity

The thermal diffusivity parameter, or thermal diffusivity coefficient, characterizes the behavior of a physical object under conditions of undetermined heat transfer and is a parameter determined experimentally in solid-state physics using an indirect method to specify thermal conductivity. Its value, through thermomechanical coupling, determines the magnitude of the loads to which a structure subjected to rapidly changing temperature may be exposed [66]. This parameter is difficult to determine for thin films due to the limitations of standard measurement methods, particularly for nonparallel sample surfaces. In physical terms, diffusivity is the ratio between the heat-transport properties and the heat storage capacity of a medium.

One of the most common methods for the determination of this parameter is the pulsed surface forcing method, which is based on the solution of a second-order Fourier differential equation with appropriate boundary and initial conditions in the sample, on which a laser pulse is applied to the front surface and the thermal response on the surface opposite to the thermal forcing is examined [42].

Despite the increasing number of modern laser techniques and non-contact temperature measurements in high-accuracy diffusometers [65], it is still quite important to prepare samples with flat parallel surfaces for testing. In accordance with the test methodology, the samples have parallel surfaces (Figure 2), and the measurement of the temperature rise is measured surface-to-surface from the side opposite to the laser forcing. Currently, there is no methodology for developing test results in software if there are uneven inclusions. Hence, unless it is possible to prepare flat test specimens, an estimate of the measurement error of such materials should be introduced. Therefore, this paper specifies the percentage error in the determination of the thermal diffusivity value of a material with known thermophysical parameters based on the example of previous experimental studies using the commercial software Comsol Multiphysics by determining the half-life.

In Parker’s pulse method, there is a simple relationship between sample thickness and half-life defined by Relation (1). For the purpose of this work, the half-life will be determined in accordance with the methodology shown in Figure 2.

The thermal diffusivity in Parker’s method is determined from the following dependence [42]:(1)$$a=\frac{1.38\cdot l^{2}}{\pi^{2}\cdot t_{0.5}}$$
where *l* is the sample thickness, t_0.5_ is the time after which half of the maximum excess temperature on the sample’s rear surface has been reached.

## 4. Numerical Model

Several geometrical models of the samples were prepared for the numerical tests. All models were designed in such a manner that the elements occurring above the parallel surfaces represent 10% of the total sample mass. Assuming that the material is homogeneous, and that the experimental tests were carried out for axially symmetrical samples, for the purposes of numerical calculations, the sample has a unit dimension in each direction. The model was assumed to have isothermal edges on the outer sides, and the laser pulse was simulated by releasing energy onto the bottom surface. As in the experimental study, the samples were covered with flake graphite to improve the absorption of the laser flash energy on the shot side as well as on the opposite side to improve the registration of the infrared thermal response on the readout side of the response in the numerical simulation. The reading took place under ideal conditions in which no additional well-conductive layers were required.

A few sample elements corresponding to 10% of the mass of the whole sample were prepared. The tested models are shown in Figure 3a–e. The geometric dimensions are given in Table 1.

Geometric models were created in the manner previously described, and with test parameter settings as below:

The subdomain settings were as follows:− General equation of heat transfer used in the model in Comsol Multiphysics:
(2)$$\rho c_{p}\frac{\partial T}{\partial t}+\nabla \left(-k\nabla T\right)=0$$

− Thermophysical properties of the tested material (Table 2):

**Table 2 materials-15-04755-t002:** Thermophysical properties of the tested material.

No	Thermophysical Parameter	Symbol	Value	Unit
1	Thermal conductivity	*k*	0.3	W/m K
2	Density	*ρ*	1500	kg/m^3^
3	Heat capacity at constant pressure	*c*	1100	J/kg K
4	Temperature	*T*(*t*_0_)	273.15	k1
5	Inward heat flux	0	100,000 (<0.01)	W/m^2^

− Element settings for temperature: Lagrange–Quadratic.

The boundary settings were as follows:

All planes were aligned symmetrically except for the bottom plane, where a laser pulse of less than 10 ms with an inward heat flux of 10^5^ W/m^2^ was simulated (Figure 4).

Computational grid settings. A calculation grid with tetrahedral elements was used (Figure 5).

## 5. Findings of Numerical Research

Numerical tests were carried out for seven different models (Table 1) with pre-determined boundary and starting conditions. In the numerical test procedure, the bottom surface corresponds to a sample aerosolized onto a flat metal plate, so it is flat, and the laser pulse was simulated on it. The temperature waveform from the nonparallel top surface was read at 0.0001 s intervals in the first temperature rise range (up to 0.02 s) and then the reading frequency was reduced to 0.0005 s.

With the models prepared in this way, half-life was checked for each model and for the parallel surfaces of the model (without the sphere part), as well as for the sphere itself. In addition, results were also given for a model with 20 per cent volume in the hemisphere element (Figure 6a,b). The obtained half-life results are included against the figures for each model. A comparison of extreme temperature increments at two points, i.e., at the parallel surface and at the highest point on the hemisphere, is also presented, in combination with the temperature change at the surface affected by the thermal forcing (Figure 6c). An exemplary model of the temperature distribution in the sample is also presented (Figure 6d). Next, the values for all the models tested are presented in a graphical form (Figure 6e) as a function of the number of hemispheres depicted as black dots on the graph. The red dots correspond to the time value t_0.5_ for the sample with 20% inclusion volume for both the sphere and the base and similarly the green color to the time value t_0.5_ for the sample with 10% share volume.

The value of the half-life for the flat-parallel sample is 13.2 ms, while each deviation from parallelism introduces an error and, depending on the number of inclusions, this parameter changes. The values of these changes are shown in the table below (Table 3).

## 6. Discussion

The presented procedure complements the determination of thermal diffusivity error for rough or uneven samples in the broad procedure of pulsed thermal testing, which further develops Parker’s method. In Parker’s method, various additional real effects accompanying the study of the heat conduction phenomenon are taken into account, except the influence of surface non-uniformity. Therefore, the methodology presented in this paper makes it possible to estimate the error of the determination of the thermal diffusivity parameter. With the additional technique of imaging the topography with, e.g., an optical profilometer, it is possible to determine an uneven volume in the sample to determine the spread of the error in the determination of the correct value.

The combination of numerical modeling and the finite element method with experimental tests and the use of a profilometer will allow a more accurate determination of thermal diffusivity values especially for materials that are problematic to manufacture with regard to the recommendations of the test methodology. Numerical measurements including surface non-uniformity can contribute to increasing the accuracy of thermal diffusivity measurements by pulse forcing methods. With this approach, it will be possible to determine the thermal properties of a material not only for perfectly manufactured samples, but for a whole range of materials that are difficult to apply in such a way as to produce perfectly flat opposite surfaces, such as paints. In previous research [36], the authors presented results assuming the parallelism of the samples [36], providing information on roughness and, as a complement to this, the potential for error due to surface irregularity. The method presented here shows the effect of bumps in the form of hemispheres whose volume corresponds to 10% (or 20%) of the volume of the sample roughness made for empirical testing. Any form of surface roughness can be used during testing once the geometric model has been implemented in the numerical calculations.

The obtained results illustrate the difference between the thermal diffusivity of the investigated material and the effective thermal diffusivity (TD), i.e., on the basis of the TD obtained from processing the measurement data of the material structure study when the structure geometry differs from the model one, the obtained results have to be interpreted as effective.

This research also provides a quantitative result. The differences in the determined diffusivity value of the resultant structure can be interpreted as actual accuracy data. The differences in the determination methodology, which involves using surface topography test data to specify the volumetric contribution of surface structures and, on this basis, determining the deviation of the test result from the true value, can be used in real-world testing.

The methodology for determining differences, which involves using the data of surface topography to specify the volume contribution of surface structures and, on this basis, determining the deviations of the test outcome from the true value, can be exploited in real testing. In this way, it is possible to determine the value of the corrections to the measurement results. In scientific works conducted based on numerical calculations of thin layers, authors have tried to simplify numerical models due to the extremely large number of computational points, which does not always translate into improvement of the obtained results, as presented in the work of Yin S et al. [67]. The obtained results are correlated with the author’s own research [36], and represent a quantitative correction for empirical data. Therefore, this paper focuses on the introduction of simplified computational models to illustrate the phenomenon of heat transfer itself and the determination of calculation errors in determining the half-life resulting from surface roughness.

In addition, the author tried to prove that the error in determining thermal diffusivity (indirectly by determining the half-life) is less than 3% when the entire volume in the non-uniform sample is replaced with the average thickness. It should be borne in mind that it is not possible to produce perfectly flat surfaces, and the roughness does not have to be uniform in accordance with the Gaussian distribution during the production of thin layers, as presented, inter alia, in the work of Z Ebrahiminejad et al. [68]. The author hopes that the methodology presented by him will be used to make computational corrections to the thermophysical parameter, thermal diffusivity, during the implementation of experimental research on similar subjects.

## 7. Conclusions

This paper presents the results of a study on the thermophysical properties of TSP coating applied by airbrush spraying using numerical replacement model studies. The error in determining half-life was determined, which translates directly into the error in determining the thermal diffusivity of the material. The numerical studies of the TSP paint were complemented by the results for TSP structures obtained by the paint casting method, which initiated the analysis of the issue. This analysis has contributed to a better understanding and determination of the thermal diffusivity of a material with a non-uniform TSP shell geometry and, at the same time, can be a methodology for testing any other material with non-planar topography. The difference in half-life determination error results for the different analyzed models can be as high as 16.7%, depending on the surface from which the responses are read on the surface opposite the impulse forcing surface. When reading the temperature from the entire rear surface, the error with the analyzed models was equal to 3.8% for the replacement model, where 10% of the volume was in one hemisphere. However, the greater the number of inclusions and the closer the surface is to the ideal sample, the smaller is the error will be. Thus, it was confirmed that this method is particularly suitable for materials that are layered on top of other materials, with the simultaneous problem of maintaining the opposite surfaces in parallel, which is crucial in determining the exact heat transfer parameters. The presented methodology as well as the test results themselves may be helpful in improving the accuracy of heat transfer tests using TSP, PSP, or other methods.

## Figures and Tables

**Figure 1 materials-15-04755-f001:**
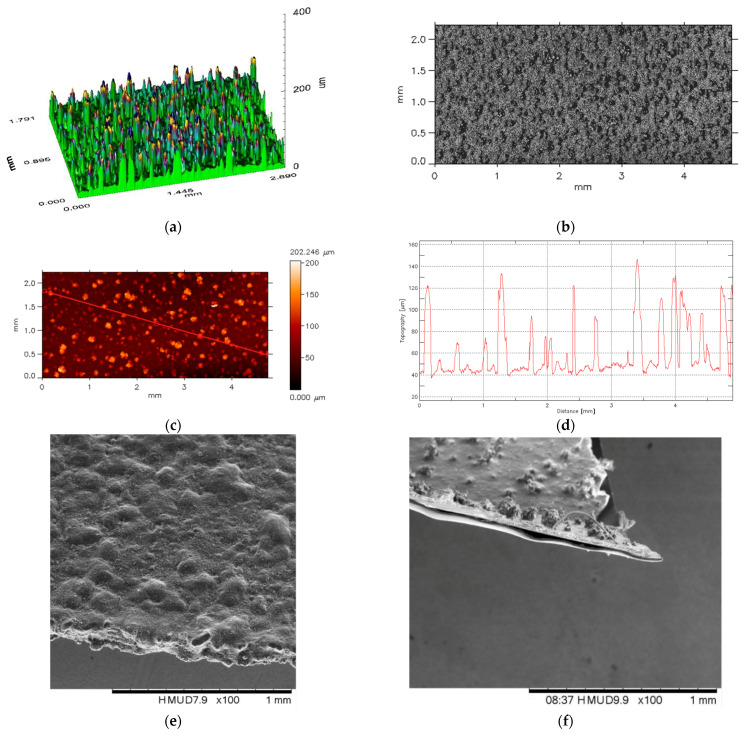
Results of the TSP coating surface morphology investigation: (**a**,**b**) three-dimensional surface image, (**c**) typical profile (red line cross-section), (**d**) diagram of a normalized dimensional spectral analysis result, (**e**,**f**) SEM images of TSP layer fragment separation for microscopic structure inspection.

**Figure 2 materials-15-04755-f002:**
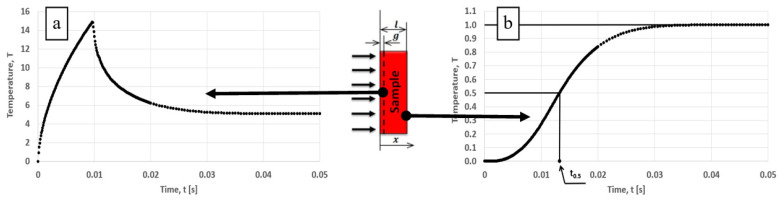
Heat transfer model for laser radiation coordinate system and initial conditions: (**a**) semi-transparent sample, (**b**) temperature rise on back surface and determination of half-life, where: g—effective sample thickness, x—Cartesian coordinate of a one-dimensional sample.

**Figure 3 materials-15-04755-f003:**
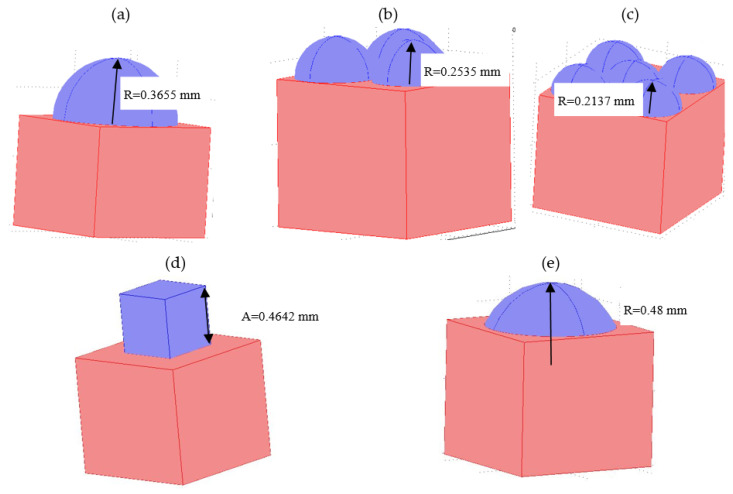
Exemplary geometrical models in the calculation program Comsol: (**a**) 1 hemisphere, (**b**) 3 hemispheres, (**c**) 5 hemispheres, (**d**) cube, (**e**) part of hemisphere.

**Figure 4 materials-15-04755-f004:**
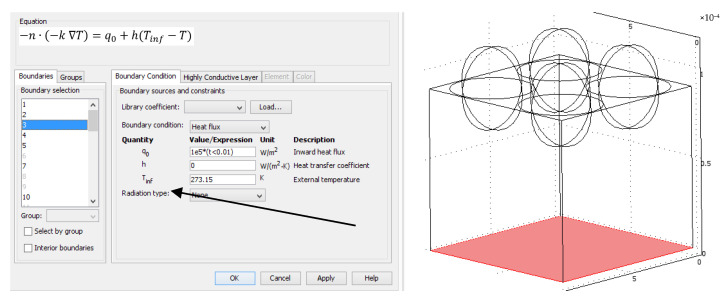
Setting the boundary conditions for an exemplary geometric model with 4 hemispheres.

**Figure 5 materials-15-04755-f005:**
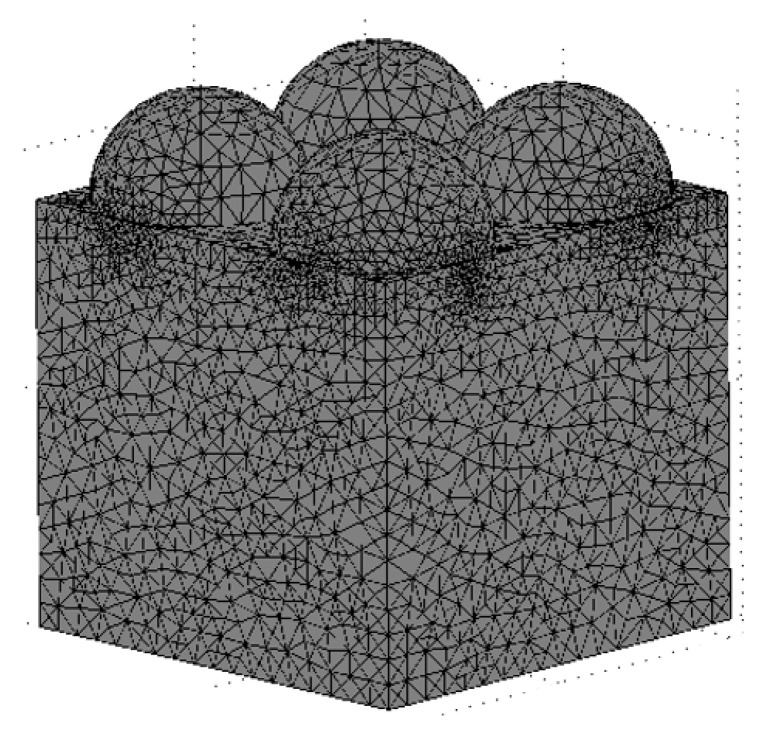
An example of a computational mesh for a geometric model with 4 hemispheres..

**Figure 6 materials-15-04755-f006:**
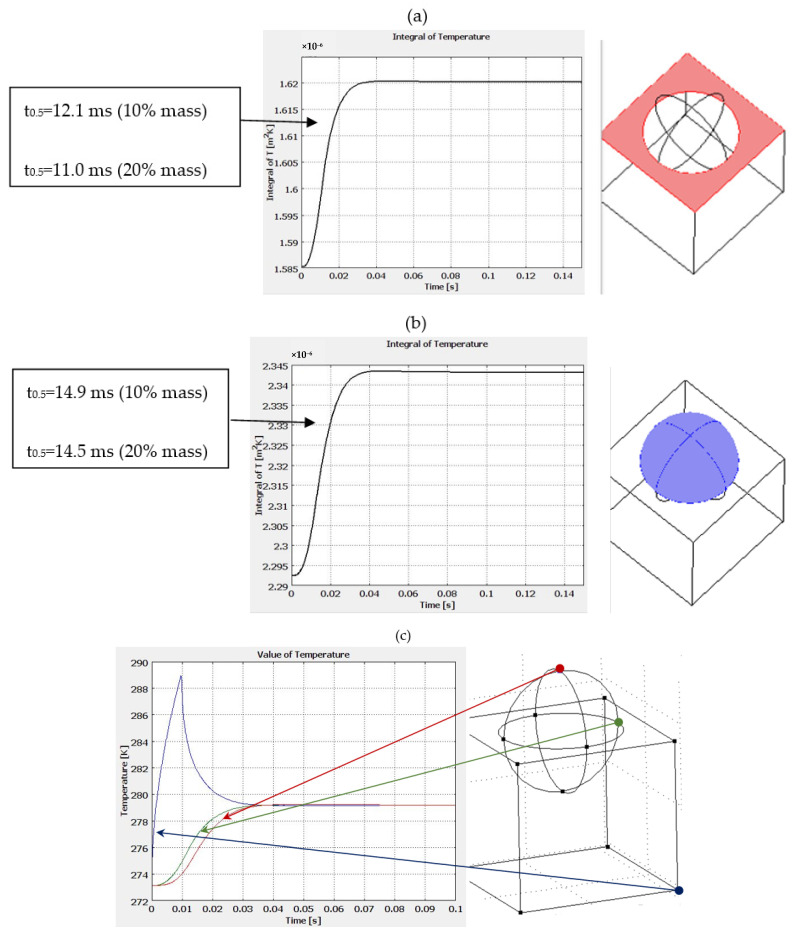
Distribution of temperature rise on the rear surface of the samples: (**a**) flat surface (without sphere), (**b**) spherical surface, (**c**) temperature distribution in selected characteristic points in the model, (**d**) example of temperature distribution in the sample after a laser shot, (**e**) half-life for different variants tested numerically.

**Table 1 materials-15-04755-t001:** Geometric dimensions of numerical models in relation to unit value.

No	Type of Inclusion	Number of Hemispheres	Radius of Hemisphere [mm]	Hemisphere Share in Total Mass [%]
1	spherical	0	0	0
2	spherical	1	0.3655	10
3	spherical	2	0.2901	10
4	spherical	3	0.2535	10
5	spherical	4	0.2300	10
6	spherical	5	0.2137	10
7	spherical	1	0.4605	20
8	spherical part	1	0.48	10
9	cube	1	0.4642—cube edge	10

**Table 3 materials-15-04755-t003:** Half-life and error of its determination for different configurations of the tested material.

Number of Hemispheres	Place of Measurement	t_0.5_ [ms]	Error [%]
0	Whole back surface	13.2	0
1	Whole back surface	13.7	3.8
2	Whole back surface	13.6	3.0
3	Whole back surface	13.4	1.5
4	Whole back surface	13.3	0.8
5	Whole back surface	13.26	0.5
1	From the surface of the hemisphere 10%	14.9	12.9
1	From the surface of the hemisphere 20%	14.5	9.8
1	From the surface around a parallel sphere 10%	12.1	8.3
1	From the surface around a parallel sphere 20%	11.0	16.7
1	Whole rear surface 20%	13.7	3.8
1	Whole back surface (inclusion cube)	12.8	3.0
1	Whole back surface (part of sphere—10%)	13.5	2.3

## Data Availability

Not applicable.

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
