# Peer review of "Effect of Surface Topology on the Apparent Thermal Diffusivity of Thin Samples at LFA Measurements"

_materials, 2022, doi:10.3390/ma15144755_

Round 1

Reviewer 1 Report

The introduction is too weak and needs updation.

The language of the research article requires rigorous revision by a native english speaker.

The novelty of the article is not well defined.

The figures used are low in resolution. The quality of figures needs to be improved.

Equations must be cited with references.

The conclusion must be quantitative. The conclusions must be derived from the major highlights of the discussions made.

Reviewer 2 Report

The paper reported the surface topology effect on the apparent thermal diffusivity of thin samples at LFA measurements. The work is suitable for the topic of the special issue. However, there are several issues should be addressed before the acceptance of this paper.

1. The abbreviation of “LFA”, “TSP”, “PSP”, etc. should give the full name at the first time.

2. As you mentioned that “determination of surface topography is particularly important”, more related work should be reviewed and discussed. For instance, 1. Fu, T., et al. (2016). Applied Physics A 122(2): 1-9.; 2. Xiang, H., et al. (2018). Ceramics International 44(9): 10376-10382.

3. Since the spherical inclusion are employed in the present work, are the results will be changed if other type of inclusion is employed?

4. The “thermophysical properties” of table 3 is improper.

Reviewer 3 Report

This manuscript presents a numerical investigation of the effect that surface topology has on the apparent thermal diffusivity of thin samples when this property is studied by instantaneous surface heat source method. Even if the study may seem a bit simple, is interesting and results are potentially useful for thin-film engineering structures. I only have the following minor remarks for authors:

1)   Authors could considerably improve the introduction of the article. Authors could widen the background by highlighting the multiple applications that thin-films could have. Avoid the use of lumping/overkilling references. See “in this film structures [5-12]”, for instance (there are more).

2)   The abstract merely describe what was done. Authors should incorporate any relevant numerical results (like authors do with the conclusions).

3)   The size of graphs axis (see Fig. 2, for instance) could be enlarged to ease data analysis.

4)   Throughout the article, time is indistinctively give with and without units. Authors should indicate when working with normalized units (if so).

5)   Authors should define “g” and “x” used in Fig. 2 (for instance).

6) Discussion is somewhat simple. Authors should deep in the physical interpretation of obtained results. More contrast with previous research could also help to improve that key section. 

Round 2

Reviewer 1 Report

The article in the present form can be accepted for publication

Author Response

Dear Reviewer,

thank you very much for your comments, of course I included them in the article and I am sending the final version

Kind regards